# Next-Generation Reconfigurable Nanoantennas and Polarization of Light

**DOI:** 10.3390/mi14061132

**Published:** 2023-05-28

**Authors:** Tannaz Farrahi, George K. Giakos

**Affiliations:** 1Department of Physics, University of Colorado, Colorado, CO 80302, USA; tafa3436@colorado.edu; 2Department of Electrical and Computer Engineering, Manhattan College, New York, NY 10463, USA

**Keywords:** polarized light, near-infrared light (NIR), optical multifunctional polarimeter, design and characterization of conjugated polymer nanomaterial films, Stokes parameters, Mueller matrix analysis, modulation, manipulation, control of polarization, reconfigurable optical nanoantennas, metasurfaces, applications

## Abstract

This study is aimed at the design, calibration, and development of a near-infrared (NIR) liquid crystal multifunctional automated optical polarimeter, which is aimed at the study and characterization of the polarimetric properties of polymer optical nanofilms. The characterization of these novel nanophotonic structures has been achieved, in terms of Mueller matrix and Stokes parameter analyses. The nanophotonic structures of this study consisted of (a) a matrix consisting of two different polymer domains, namely polybutadiene (PB) and polystyrene (PS), functionalized with gold nanoparticles; (b) cast and annealed Poly (styrene-b-methyl methacrylate) (PS-PMMA) diblock copolymers; (c) a matrix of a block copolymer (BCP) domain, PS-b-PMMA or Poly (styrene-block-methy methacrylate), functionalized with gold nanoparticles; and (d) different thicknesses of PS-b-P2VP diblock copolymer functionalized with gold nanoparticles. In all cases, backscattered infrared light was studied and related to the polarization figures-of-merit (FOM). The outcome of this study indicates that functionalized polymer nanomaterials, depending upon their structure and composition, exhibit promising optical characteristics, modulating and manipulating the polarimetric properties of light. The fabrication of technologically useful, tunable, conjugated polymer blends with an optimized refractive index, shape, size, spatial orientation, and arrangement would lead to the development of new nanoantennas and metasurfaces.

## 1. Introduction

Progress in nanofabrication and the characterization of optical nanostructures have led to the proliferation of optical nanodevices, such as optical nanoantennas, for a wide range of applications, such as sensors, quantum detection, photovoltaic systems, biological and chemical sensing and detection, efficient light transmitters and control, optical imaging and computer visualization, and the nanofabrication of photonic chips, with emerging applications in optical communications and quantum computing. Optical nanoantennas [1,2,3,4,5,6,7,8,9,10,11,12,13,14,15,16,17,18,19,20] have similar functionality to that of radiofrequency (RF) antennas, although these two classes of antennas exhibit drastic differences in terms of their physical principles, material properties, fabrication, excitation features, and scale engineering. Nanoantennas can be classified into metallic (plasmonic nature) nanoantennas, dielectric nanoantennas and metal-dielectric nanoantennas [5]. Plasmonic nanoantennas are based on the interaction between localized surface plasmon resonances with metal nanoparticles, typically gold nanoparticles (AuNPs). The efficiency of the antenna is affected by the material from which it is made. Dielectrics nanoantennas are typically fabricated from transparent, high refractive index materials such as silicon or germanium, which provide good optical properties and minimize light scattering. Metal-dielectric nanoantennas offer a compromise of both mentioned types; although metal-dielectric nanoantennas would provide a significant Purcell factor, high scattering directionality, and low scattering losses, their design complexity necessitates careful consideration for the realization of optimal design parameters [1,5,6].

Gold nanoparticles (AuNPs) offer distinct advantages because of their chemical stability, biocompatibility, ease of synthesis and functionalization, and enhanced optical and electronic tunability. As a result, gold and silver nanoparticle monolayer substrates offer unique opportunities for surface plasmonic studies. Metal nanoparticles have various unusual chemical and physical properties which make them attractive for applications such as optics, sensing, electronics, chemistry, security, medicine, and biology. By incorporating both polymers and AuNPs into films and devices, increased tunability, manipulation, and reconfigurability of the optical and electrical properties would result. The doping of polymers with metal nanoparticles such as gold enhances the conductivity of the polymers [21,22,23,24]. Moreover, the refractive index of polymers can be tuned by adding high refractive index nanoparticles, although large-size gold nanoparticles introduce losses. In addition, the good control of feature sizes and morphology, and reliable fabrication procedures, is imperative. Functionalized polymers with gold nanoparticles (AuNPs) can provide enhanced multifunctionality such as (a) the control of interparticle spacing which allows the modulation of the film properties; (b) chemical and biochemical functionality which can facilitate interaction with analytes and other environmental modifiers; (c) reinforced films through crosslinking between the components; (d) the ability to control light at the nano/micro level, increased sensitivity in sensing, and the production of small form factor, lightweight nanodevices; and (e) significant improvement in electrical, thermal, mechanical, physical, and surface properties. Similarly, biological macromolecules have been used to build defined nanostructures. Among the biological macromolecules, DNA is one of the most interesting polymer templates because of its diameter of only 2 nm and the micrometer-long distribution of well-defined sequences of DNA bases. Several papers reported the fabrication of a one-dimensional arrangement templated by DNA [25,26,27,28,29,30,31,32]. Other biological materials such as peptides, viruses, lipids, and biopolymers were also used [33,34,35,36]. The most important advantage of using biological materials is their single molecular weights, which provide a controlled length of one-dimensional arrays. Nonbiological templates such as carbon nanotubes and the polycation-molecule-templated self-assembly of gold nanoparticles have also been reported [37,38,39].

The motivation of this study stems from the fact that plasmonic nanoantennas exhibit limited dynamic tunability due to the fixed permittivity of conventional metals. On the other hand, polymer thin films are finding potential applications in emerging areas of nanophotonics such as solar energy cells as well as in nanoelectronics such as patterned active layers for transistors or as sensors for biomedical applications. By controlling the index of refraction of nanostructured polymer films, as well as the concentration and shape of the embedded nanoparticles in conjunction with the thickness of the polymer film, enhanced tunability would result. The realization of technologically useful conjugated polymer materials depends not only on the quality of the nanoparticles (e.g., size and shape) but also on their spatial orientation and arrangement. On the other hand, nano-sized irregularities may lead to enhanced optical loss. The fabrication and engineering of plasmonic and electronic structures using novel directed-assembly-controlled metallic nanoparticle ordering in polymer blend thin films would yield novel tunable nanoantennas designs and metasurfaces. The electronic structure of the polymer chain strongly influences the characteristics of the embedded metal nanoparticles. The building and patterning of the metal nanoparticles into organized structures is a potential route to chemical, optical, magnetic, and electronic devices with useful properties. For instance, a repetitive metal-dielectric structure can be produced by the co-assembly of diblock copolymers and gold nanoparticles. This structure is suitable for the fabrication of photonic bandgap materials for near-infrared or optical frequencies. Dielectric contrast can be further increased by the selective isolation of gold nanoparticles within one domain among adjacent block copolymer microdomains. The resulting photonic structure has high reflectivity in the visible and NIR region which indicates that absorption losses can be reduced in periodic microstructures [38].

Butadiene is an attractive monomer that is conveniently available from steam cracking. It can be incorporated into polymers in different fashions (1,2 or vinylic incorporation, 1,4-*cis* and 1,4-*trans* incorporation), which results in a wide variety of different polymer microstructures differing in crystallinity and mechanical as well as thermal properties [40,41]. Most commonly, polybutadiene (PB) is known for its elastomeric properties. Polystyrene (PS) is a commonly used aromatic polymer used to make many distinct types of plastics due to its thermoplastic attributes. Polystyrene (PS) is a brittle plastic with poor strength. It is a clear thermoplastic commonly used in disposable plastic utensils and Styrofoam and utilized in many commercial sectors, from automobile parts to laboratory ware. While it is non-biodegradable, it is of interest because of its affordable and long-lasting qualities. PS has a glass transition temperature of around 100 °C, much higher than PB. The blend of these two polymers is partially miscible in that they have an accessible phase miscibility at low molecular weights, but they are generally immiscible for higher molecular weights. When gold nanoparticles are mixed with a matrix of two different polymers such as polybutadiene (hydrophilic) and polystyrene (hydrophobic), the accumulation of the gold nanoparticles is expected to occur in both the polymer-rich phase and at the interface separating the domains of PS and PB. Polymethyl methacrylate (PMMA) is a low-cost polymer with exceptional qualities such as transparency, thermal stability, electrical insulation, and mechanical resistance. Due to its low cost and high output, PMMA has sparked significant interest for use as optical components and in optoelectronic devices. Most polymers are immiscible unless there are specific interactions; therefore, coarse phase separation often occurs. Diblock copolymers bypass this limitation, since the scale of the domains is restricted to the sizes of the individual homopolymer, typically on a scale of tens of nm. Another advantage is that the size of different blocks can be modified by varying the concentration of the different components. As a result, enhanced tunability of the physical, mechanical, and optical parameters can be achieved. Enhanced tunability would result by controlling the index of refraction of the nanostructured polymer film, either by controlling the concentration or the shape of the embedded nanoparticles in conjunction with the thickness of the polymer film. 

Polarimetric characterization, visualization, and sensing offers unique advantages for a wide range of detection and classification problems due to its intrinsic potential for high contrast and a dynamic range in different technological areas, spanning remote sensing, material characterization, industrial inspection, and medical imaging. It offers unique optical, chemical, mechanical, morphological, biological, and metabolic information regarding material composition and surface characteristics, including the underlying physical mechanisms [42,43,44,45,46,47,48,49,50,51,52,53,54]. Flat optics encompasses nanostructures consisting of nanometer-size-spaced optical scatterers—also known as metasurfaces or meta-optics [4,12,55,56,57,58,59,60,61,62,63,64,65,66,67,68,69,70,71,72,73]. Metasurfaces, are startling technological advancements because of their potential to offer enhanced tunability and the manipulation of various free degree-of-freedom light parameters such as angle of incidence; wavelength; polarization; phase, yielding structured light within a compact footprint; unsurpassable versatility; tunability; and reconfigurability. Referring to their multifunctional potential to tune the polarization properties of input light, metasurface materials can lead to numerous applications, reducing the function of many polarizers and waveplates into a single optical component that can be integrated into several optical devices and instruments. As a result, it is possible to realize polarization-component-free polarized light sources, therefore eliminating problems associated with extra weight, cost, and energy loss, yielding the development of efficient nanoantennas with applications in adaptive computer visualization systems, microscopes, portable and wearable devices, future optical communications, and imaging and sensing systems [3,4,12,14,15,17,63,64,65,66,67,68,69,70,71,72,73]. 

The purpose of this study is to present the design, calibration, and development of a near-infrared (NIR) liquid crystal multifunctional automated optical polarimeter, aimed at the study and characterization of the polarimetric properties of functionalized polymer optical nanofilms. The experiments were performed under backscattering geometry using a laser beam operating at 1065 nm [40]. The nanophotonic structures of this study consisted of (a) a matrix consisting of two different polymer domains, namely polybutadiene (PB) and polystyrene (PS), functionalized with gold nanoparticles; (b) cast and annealed Poly (styrene-b-methyl methacrylate) (PS-PMMA) diblock copolymers; (c) a matrix of a block copolymer (BCP) domain, PS-b-PMMA or Poly (styrene-block-methyl methacrylate), functionalized with gold nanoparticles; and (d) different thicknesses of PS-b-P2VP diblock copolymer (Polystyrene-b-poly(2-vinylpyridine), functionalized with gold nanoparticle thin films. The characterization of these novel nanophotonic structures has been achieved, in terms of Mueller matrix and Stokes parameter analysis [42,43,44,45,46,47,48,49,50,51,52,53,54].

## 2. Polarimetric Formalism

The calibration of the automated electro-optical polarimetric system, including the acquired experimental data presented in this study, were analyzed and processed following the algorithms and procedures reported in [40,45,46,50,51].

The linear depolarization ratio (LDR) is the ratio expressed as the ratio of the two linearly polarized optical powers incident on the receiver when the source is also linearly polarized, namely
(1)$$LDR=\frac{I_{⊥\text{\_}sample}}{I_{∥\text{\_}sample}}.$$

Here, $I_{∥\text{\_}sample}$ is the intensity of the co-polarized state where maximum intensity is obtained and $I_{⊥\text{\_}sample}$ is the intensity of the cross-polarized state where the minimum intensity is obtained. Therefore, the concept of residual intensity R.I. was utilized, expressed in this study as an attempt to quantify the net polarized backscattered intensities, as
(2)$$R.I.=\left|I_{⊥\text{\_}sample}-I_{∥\text{\_}sample}\right|.$$

The degree of linear polarization (DOLP) is another important optical parameter that can be used to study different samples and differentiate them. Its values are always between 0 and 1 and it is given by
(3)$$DOLP=\frac{\left|I_{⊥\text{\_}sample}-I_{∥\text{\_}sample}\right|}{\left|I_{⊥\text{\_}sample}+I_{∥\text{\_}sample}\right|}=\frac{R.I.}{\left|I_{⊥\text{\_}sample}+I_{∥\text{\_}sample}\right|}.$$

The polarization state of any beam of light can be completely described by a column matrix of order four called the Stokes vector. The elements of the Stokes vector are called the Stokes parameters and represent the various intensity measurements for different configurations of the input polarization state. The Stokes vector represented by *S* is thus given by Equation (4):(4)$$S=\left[\begin{array}{c} \begin{array}{c} S_{0}\\ S_{1} \end{array}\\ \begin{array}{c} S_{2}\\ S_{3} \end{array} \end{array}\right]=\left[\begin{array}{c} \begin{array}{c} I_{H}+I_{V}\\ I_{H}-I_{V} \end{array}\\ \begin{array}{c} I_{P}-I_{M}\\ I_{R}-I_{L} \end{array} \end{array}\right]$$

The first element, *S*_0_, is the total intensity of light given by the sum of the linear horizontal and vertical intensities. *S*_1_ is the amount of linear horizontal or vertical polarization, given by the difference between the corresponding intensities. *S*_2_ is the amount of linear ±45° polarization, given by the difference between their intensities. *S*_3_ is the amount of right or left circular polarization present in the beam, given by the difference between the respective intensities. The Stokes parameters are real quantities that can be measured directly and have units of watts per meter squared (W/m^2^).

The Mueller matrix *M* for an optical system is a 4 × 4 array that maps the transformation between the Stokes vector *S* incident on an object and the vector *S*′ that is transmitted or deflected or scattered from the object. The Mueller matrix also captures information about all the optical components that are present in the system between the incident and the transformed vector. This means that each of the components can be individually characterized by their own Mueller matrices *M_i_*’s and the resulting matrix M can be expressed as a product of all the individual Mueller matrices. This implies that the contribution of each component to the system can be clearly accounted for and the addition or removal of any component to or from the system can be exactly mathematically represented by Equations (5) and (6).
(5)$$S^{\prime }=\left[\begin{array}{c} \begin{array}{c} {S_{0}}^{\prime }\\ {S_{1}}^{\prime } \end{array}\\ \begin{array}{c} {S_{2}}^{\prime }\\ {S_{3}}^{\prime } \end{array} \end{array}\right]=\left[\begin{array}{cc} \begin{array}{cc} m_{11} & m_{12}\\ m_{21} & m_{22} \end{array} & \begin{array}{cc} m_{13} & m_{14}\\ m_{23} & m_{24} \end{array}\\ \begin{array}{cc} m_{31} & m_{32}\\ m_{41} & m_{42} \end{array} & \begin{array}{cc} m_{33} & m_{34}\\ m_{43} & m_{44} \end{array} \end{array}\right]\left[\begin{array}{c} \begin{array}{c} S_{0}\\ S_{1} \end{array}\\ \begin{array}{c} S_{2}\\ S_{3} \end{array} \end{array}\right]=M\cdot S$$
(6)$$M=M_{1}.M_{2}.\;M_{3}\ldots .M_{n}$$

There are numerous ways of measuring the Stokes parameters of a beam of light and the most common is the Fourier analysis method that uses a rotating quarter-wave retarder to obtain the Fourier coefficients. The biggest advantage of this method is that the Stokes vector can be directly obtained from the Fourier coefficients. The method of obtaining the Stokes parameters from the Fourier coefficients of the optical system is described below in detail. To obtain the coefficients, the total Mueller matrix must be derived from the various components of the system. The experimental setup of the system shows that the Stokes vector of the beam scattered from the sample is affected by the rotating retarder and the linear −45° polarizer. The scattered beam in turn carries information about the effect of the optical activities of the biological samples on the Stokes vector of the input laser beam. The Mueller matrices of the rotating quarter wave retarder and the analyzer polarizer are, respectively, given by Equations (7) and (8):(7)$$M_{retarder\;}=\left[\begin{array}{cccc} 1 & 0 & 0 & 0\\ 0 & Cos^{2}2\theta & Sin2\theta \;Cos2\theta & -Sin2\theta \\ 0 & Sin2\theta \;Cos2\theta & Sin^{2}2\theta & Cos2\theta \\ 0 & Sin2\theta & -Cos2\theta & 0 \end{array}\right]$$
(8)$$M_{polarizer}=\frac{1}{2}\left[\begin{array}{cccc} 1 & 0 & -1 & 0\\ 0 & 0 & 0 & 0\\ -1 & 0 & 1 & 0\\ 0 & 0 & 0 & 0 \end{array}\right]$$

The Stokes vector of the scattered beam from the sample is assumed to be *S*. The output Stokes vector *S*′ is then given by Equation (9) below, and its total intensity is the *S*_0′_ element given by Equations (10) and (11) as
(9)$$S^{\prime }=M_{polarizer}.M_{retarder}.S$$
(10)$$S^{\prime }=\frac{1}{2}\;\left(S_{0}-S_{1}Sin2\theta \;Cos2\theta -S_{2}Sin^{2}2\theta -S_{3}Cos2\theta \right)\left[\begin{array}{c} \begin{array}{c} 1\\ 0 \end{array}\\ \begin{array}{c} -1\\ 0 \end{array} \end{array}\right]$$
(11)$$S_{0}^{\prime }=I\left(\theta \right)=\frac{1}{2}\;\left(S_{0}-S_{1}Sin2\theta \;Cos2\theta -S_{2}Sin^{2}2\theta -S_{3}Cos2\theta \right)$$

Rewriting the intensity expression *I*(*θ*) in terms of its trigonometric half-angle formula reduces it to the standard form of the truncated Fourier series given in Equation (12). It is seen from this expression that the output intensity is described by four Fourier coefficients, namely *A* (the DC component), *B*, *C*, and *D* (the frequency harmonics). The equations defining these coefficients are given by Equations (13)–(16). Furthermore, it is seen that the Stokes parameters can be easily obtained from the coefficients as shown in Equations (17)–(20).
(12)$$I\left(\theta \right)=\frac{1}{2}\;\left[\left(S_{0}-\frac{S_{2}}{2}\right)-S_{3}Cos2\theta +\frac{S_{2}}{2}Cos4\theta -\frac{S_{1}}{2}Sin4\theta \right]$$
(13)$$A=\left(S_{0}-\frac{S_{2}}{2}\right)=\frac{1}{8}\overset{8}{\underset{n=1}{\sum }}I\left(\frac{n\pi }{8}\right)$$
(14)$$B=S_{3}=\frac{1}{4}\overset{8}{\underset{n=1}{\sum }}I\left(\frac{n\pi }{8}\right)Cos\left(\frac{n\pi }{4}\right)$$
(15)$$C=\frac{S_{2}}{2}=\frac{1}{4}\overset{8}{\underset{n=1}{\sum }}I\left(\frac{n\pi }{8}\right)Cos\left(\frac{n\pi }{2}\right)$$
(16)$$D=\frac{S_{1}}{2}=\frac{1}{4}\overset{8}{\underset{n=1}{\sum }}I\left(\frac{n\pi }{8}\right)Sin\left(\frac{n\pi }{2}\right)$$
(17)$$S_{0}=A+C$$
(18)$$S_{1}=-2D$$
(19)$$S_{2}=2C$$
(20)$$S_{3}=-B$$

The Stokes parameters obtained were used to estimate polarimetric figure-of-merits (FOM) used to process the acquired experimental data, namely the following:(21)$$\mathrm{Degree}\;\mathrm{of}\;\mathrm{Polarization}:\;DOP=\frac{\sqrt{S_{1}^{2}+S_{2}^{2}+S_{3}^{2}}}{S_{0}}$$
(22)$$\mathrm{Degree}\;\mathrm{of}\;\mathrm{Linear}\;\mathrm{Polarization}:\;DOLP=\frac{\sqrt{S_{1}^{2}+S_{2}^{2}}}{S_{0}}$$
(23)$$\mathrm{Degree}\;\mathrm{of}\;\mathrm{Circular}\;\mathrm{Polarization}:\;DOCP=\frac{S_{3}}{S_{0}}$$

A detailed analysis of the object can be obtained by decomposing the Mueller matrix into three matrices, namely the depolarization matrix, retardance matrix, and diattenuation matrix [44]. Since the presented study involves angular backscattering polarimetric measurements from the object, in terms of the aspect angle of an object, *θ*, it yields, the Mueller matrix can be decomposed according to [45]
(24)$$M\left(\theta \right)=M_{depol}\left(\theta \right)M_{ret}\left(\theta \right)M_{diat}\left(\theta \right)$$
where $M_{depol}\left(\theta \right)$ accounts for the depolarizing effects of the medium, $M_{ret}\left(\theta \right)$ accounts for the linear birenfringence and optical activity, and $M_{diat}\left(\theta \right)$ describes the effects of linear and circular dichroism. From these matrices the diattenuation, retardance, and depolarization characteristics of the medium are readily determined.

The depolarization is quantified in terms of the depolarization index, *P_D_*, according to
(25)$$P_{D}\left(\theta \right)=Dep\left(M\left(\theta \right)\right)=1-\frac{\sqrt{\left(\sum_{i,j}m_{ij}^{2}\left(\theta \right)\right)-m_{00}^{2}\left(\theta \right)}}{\sqrt{3}m_{00}^{2}\left(\theta \right)}$$
where *m_ij_*(*θ*) are the Mueller matrix elements as a function of the aspect angle. From the decomposed retardance matrix, $M_{ret}\left(\theta \right)$, the total retardance, *R*, which includes the effects of both linear and circular birefringence, can be expressed as
(26)$$R\left(\theta \right)=\cos^{-1}\left(\frac{tr\left(M_{ret}\left(\theta \right)\right)}{2}-1\right)$$
where $tr\left(M_{ret}\left(\theta \right)\right)$ is the trace of the retardance matrix. The diattenuation, *d*, is dependent of the first-row vector of the Mueller matrix. This vector describes differential attenuation for both linear and circular polarization states and can be expressed as
(27)$$d\left(\theta \right)=\frac{1}{m_{00}\left(\theta \right)}\times \sqrt{m_{01}^{2}\left(\theta \right)+m_{02}^{2}\left(\theta \right)+m_{03}^{2}\left(\theta \right)}$$

It is well known that depolarization expresses the loss of the original polarization of light after interaction with the sample material. Diattenuation indicates the absorption of a specific polarization of light, while retardance is the phase shift introduced to a specific polarization of light after interaction with the sample material. Overall, optical nanofilms exhibiting reduced depolarization, retardance, and diattenuation are highly desirable.

## 3. Instrumentation Design

### 3.1. Polarimetric System Descriprion

The experiment was performed using electronically controlled liquid crystal (LC) retarders and rotators, as shown in Figure 1. These LC optical devices were used to control the polarization states of the beam electronically, by means of an applied voltage. With no voltage applied, the LC molecules lay parallel to the surface and maximum retardance was achieved. As the voltage was applied across the liquid layer, the molecules rotated towards the direction of the applied electric field, which introduced retardance such that different polarization states were achieved. The liquid crystal devices included a controller, which helped to provide a square wave with a changing amplitude that adjusted the retardation. The controller also included a driver and small VI’s to interface the devices with LabVIEW. The calibration was performed via a two-step procedure.

The imaging system contained two branches: a polarization-generating branch and a polarization-analyzing branch. Light from a 1065 nm laser source operating at a pulse repetition rate (PRR) of 200 Hz was sent through the polarization-generating branch, consisting of a linear polarizer P1 set at +45°, a liquid crystal rotator with fast axis at 135°, and a liquid crystal retarder with fast axis at 45°, so that that linearly polarized waves maintained their initial polarization state.

The liquid crystal (LC) multifunctional polarimetric imaging platform was capable of deriving the full 16 element Mueller matrix (MM) of an object using liquid crystal (LC) devices. The system was highly automated by utilizing NI Labview software to control the devices and measurement states and perform a detailed calibration of the optical system. It was fully reconfigurable and scalable, providing enhanced multifunctional surveillance capabilities through dedicated spatial, timing and spectral modules, supported by advanced calibration and image pre-processing and post-processing techniques developed in-house.

### 3.2. Voltage Calibration of Liquid Crystal Retarders and Rotators

The polarimetric platform was first aligned. The calibrated voltages for the rotator and the retarder in the generator arm were then obtained. This was achieved by first calibrating the generator arm and then calibrating the analyzer arm. The calibration was obtained by adding each component one by one. Appropriate rotation or retardation for all the states was achieved by applying voltages to the devices. The calibrated states were the different polarization states: Horizontal (H), Vertical (V), +45° linear (P), −45° linear (M), Right circular (RCP), and Left circular (LCP).

Calibration was obtained by using a method called null-intensity method where the analyzer polarizer was always kept cross-polarized to the generator polarizer and a voltage was applied to each of the LC devices such that the null intensity was obtained [42,48]. Because of the presence of some system noise in the experiment, the true null was difficult to observe using this method for alignment. Thus, a different method was used for calibration. For this, the analyzer polarizer was rotated by a small angle Δθ to measure the intensities at 90° + Δθ and 90° − Δθ. When these two measurements were equal, the null exactly midway between these angles was the calibration voltage. The measurements were repeated until the two equal measurements were obtained. This method is known as the method of swings [48]. The calibration setup for the system was as shown in Figure 2.

### 3.3. Calibration Using Known Targets with Known Mueller Matrix (Accuracy Test)

After the liquid crystal devices (retarders and rotators) were calibrated, these calibration voltages were first tested using some components of known Mueller matrix. The components used were air, linear horizontal polarizer (LHP), i.e., a polarizer having its transmission axis in the horizontal direction, and linear vertical polarizer, i.e., a polarizer having its transmission axis in the vertical direction. These test experiments were conducted in transmission mode, i.e., placing the generator and the analyzer arm in the line-of-sight. Between the generator and the analyzer arm, the object to be tested was placed. For air, no object was needed. The polarizers (linear horizontal and vertical) used for the experiment were specified for the wavelength range between 700 nm and 1100 nm, which is within the range of the laser that was being used (1065 nm). The experimental setup for the test was similar to that one shown in Figure 2 except that the object to be tested was inserted between the generator arm and the analyzer arm. The plots of the ideal and experimentally obtained Mueller matrix intensities of three calibration targets are plotted in Figure 3, Figure 4 and Figure 5.

A statistical analysis of the measured and ideal Mueller matrix (MM) elements for air, LHP, and LVP is tabulated in Table 1.

From the statistical analysis, the standard deviation for the Mueller matrix of air, LHP, and LVP obtained were 0.114, 0.0478, and 0.0364, respectively, which showed high the degree of accuracy of the measurements, showing that the liquid crystal devices had been calibrated accurately.

## 4. Nanofabrication of Polymer Nanomaterials

(1)Thin films of polystyrene (PS) and polybutadiene (PB) domains functionalized with gold nanoparticles

Polystyrene (PS) [20] with an average relative molecular mass of Mw = 3000 g/mol and a polydispersity index (ratio of mass average to number-average relative molecular mass, Mw/Mn) of 1.09, and polybutadiene (PB) with a Mw = 2400 g/mol and a polydispersity index of 1.05, from Polymer Source Inc., were used for our experiments. Laboratory grade toluene was used as the solvent for polymers and was purchased from BDH chemicals. Homopolymer solutions of PS and PB in toluene were prepared separately by stirring the polymers in the solvent overnight. Blend solutions were made by dissolving the 50:50 PS:PB in the common solvent toluene, such that the total polymer concentration in the solvent was 3% by mass, with a 1% concentration of 5 nm gold nanoparticles. The solutions were then stirred overnight and cast as thin films on a silicon substrate by spin coating at a spin speed of 2000 rpm [40].

In addition, the polarimetric properties of the following three block copolymers were studied, namely
(2)Poly (styrene-b-methyl methacrylate) PS-PMMA used as cast/annealed.

The PS-PMMA copolymer [20], which contains no gold nanoparticles and is a pure block copolymer was studied both as-cast and annealed [40]. Specifically, Poly (styrene-b-methyl methacrylate) (PS-PMMA) lamellae diblock copolymer with a polydispersity index (PDI) = 1.18 was purchased from Polymer Source Inc. and was dissolved in toluene (3%). Lamellar thin films (thickness = 183 nm) with a molecular weight of 260 Kg/mol^−1^ and 289 Kg/mol^−1^Ps-b-PMMA were flow-cast on ultraviolet–ozone (UVO)-cleaned silicon substrates which were purchased from Silicon Quest International, Inc. Film thicknesses were determined using a thin film interferometer (F-20 UV Thin Film Analyzer, Filmetrics, Inc., San Diego, CA, USA) with a resolution of 1 nm. To induce microphase separation, PS-PMMA films were annealed at 180 °C for 3 h. (270-b-289 kg/mol. As a result, 183 nm PS-PMMA cast and annealed samples were fabricated and used throughout this study. In the processes of fabricating the polymer samples, thermal annealing was used to induce micro phase separation. In this study, we used atomic force microscopy (AFM) to find the orientation of polymer blocks within the casted samples and to observe the separation of micro phases. The technique applied in this study is called tapping mode atomic force microscopy. In this method, a sharp tip (probe), which is usually a silicon nitride crystal or a silicon, oscillates above the specimen surface. The experimental arrangement depicting the AFM process is shown in Figure 6.

While the probe scans the surface of the sample, the tip will rise and fall on the surface with different features. The laser beam, which is pointed at the probe, will be reflected to a sensor. As the probe goes up and down, the laser beam will hit different portions of the sample, and consequently, it will hit different sections of the sensor. The oscillation frequency of the probe will change according to the harder or softer features of the sample surface and will generate the phase contrast images. The AFM measurements were mostly completed at room temperature and in air on films cast on flat wafers such as silicon.

The surface topography of the sample nanocomposites was imaged in this study using a Dimension Icon atomic force microscope (AFM) (Bruker AXS) in the Peak Force Quantitative Nanomechanical Property mapping mode [40]. Pre-annealing and post-annealing backscattered measurements were completed for the samples. AFM measurements were performed on the oven-annealed sample with the purpose of phase separation. AFM measurements of the as-cast sampleshow that the film is smooth without phase separation structures. An AFM measurement of the annealed sample is provided in Figure 7. It is observed that the sample surface is exceptionally smooth, while the lack of any distinguishable features indicates the parallel orientation of lamellae.
(3)PS-PMMA and PS-PMMA/AuNP

The copolymers for this study were purchased from Polymer Source Inc., Dorval, QC, Canada with the following properties, and were used as obtained [40]. The first block copolymer was PS-b-PMMA or Poly (styrene-block-methyl methacrylate) with a molecular weight of PS $57\;\mathrm{kg}/{\mathrm{mol}}^{-1}$, PMMA $25\;\mathrm{kg}/{\mathrm{mol}}^{-1}$. Thiol-terminated polystyrene (PS-SH) Au NPs with an average core diameter <d> ≈3.61±1.29 nm and a degree of polymerization of surface-grafted chains of N=9 were synthesized via the phase transfer reduction of [$AuCl_{4}$] in the presence of the thiol ligands. An appropriate amount of AuNPs were mixed with the PS-PMMA solutions. The concentration of PS-b-PMMA in volume was 3%. The concentration of Au NPs with respect to PS-b-PMMA was 10%. Thin films (120 nm and 180 nm) were flow-coated on ultraviolet-zone-cleaned silicon substrates. The surface topography was imaged using a Dimension Icon atomic force microscope (Bruker AXS) in the Peak Force Quantitative Nanomechanical Property mapping mode.

From the above-mentioned block copolymer, three samples were fabricated over bare silicon in different thicknesses, listed as follows:(1)PS-b-PMMA 180 nm(2)PS-b-PMMA/Au NP 180 nm(3)PS-b-PMMA/Au NP 120 nm(4)PS-P2VP (Polystyrene-b-poly)2-vinylpyridine//AuNP

The PS-P2VP/AuNP copolymer is the poly (styrene-*b*-2-vinylpyridine) PS-*b*-P2VP diblock copolymer. This is a typical hydrophobic–hydrophilic BCP with both blocks being insoluble in water and soluble in organic solvents. The chemical structures of the P2VP and PS are shown in Figure 8. The PS-b-P2VP diblock copolymer used in this study had a molecular weight of PS $40\;\mathrm{kg}/{\mathrm{mol}}^{-1}$, P2VP $18\;\mathrm{kg}/{\mathrm{mol}}^{-1}$. Total Thiol-terminated polystyrene (PS-SH) Au NPs with an average core diameter of <d> ≈6.8 nm were synthesized via the phase transfer reduction of [$AuCl_{4}$] in the presence of the thiol ligand PS (red section).

The concentration of Au NPs with respect to PS-b-PMMA2VP was 6%. The samples made from this block copolymer with different thicknesses are listed as the following:
(1)PS-P2VP/Au NPs 105 nm(2)PS-P2VP/Au NPs 40 nm

Note that 105 nm and 40 nm are the thicknesses of the samples. After the initial experiments on the as-cast samples, they were vacuum-oven annealed for 1 h at 180 °C. Optical experiments were repeated for the annealed samples. AFM measurements were completed for the 40 nm PS-P2VP/Au NP annealed sample and the 105 nm PS-P2VP/Au NP annealed sample and they are provided in Figure 9 and Figure 10, respectively.

## 5. Experimental Results and Discussion

The accuracy of the calibration voltages was verified by calculating the Mueller matrix of known test objects and comparing those results with the ideal ones. The plots of Figure 3, Figure 4 and Figure 5, including the statistical analysis of the MM elements of Table 1, clearly indicate an excellent agreement between theory and experiment for the selected known test objects. As a result, the liquid crystal retarders and rotators were calibrated accurately at different polarization states and hence could be used for the experimental analysis of the polymer nanofilms. Using the experimental arrangement of Figure 1, repeatability experiments took place aimed at assessing the stability of the polarimetric system, as shown in Figure 11. The stability of the polarimetric system was assessed by performing a set of six repeated measurements of co-polarized detected intensities over a time of 8.04 min. By inspecting Figure 11, it is observed that the polarimetric system exhibited excellent stability over time.

The Mueller matrices of conjugated polystyrene (PS) and polybutadiene (PB) blended with Au NPs, and plain silicon substrate were determined. The Mueller matrices of conjugated polystyrene (PS) and polybutadiene (PB) blended with Au NPs, and plain silicon substrate are shown in Figure 12. By applying the Mueller matrix decomposition technique (Equation (24)) [45], in conjunction with Equations (25)–(27), the depolarization index, diattenuation, and total retardance matrices for the plane silicon and the functionalized polystyrene (PS) and polybutadiene (PB) domains were estimated and plotted in Figure 13, Figure 14 and Figure 15, respectively. The plots of Figure 13a–c indicate that thin films of functionalized polystyrene (PS) and polybutadiene (PB) domains with gold nanoparticles depolarize incident light less than the silicon substrate, while they exhibit higher retardance and less diattenuation than silicon. Uncoated surfaces such as plain silicon act as diattenuators; in contrast, coated films on the surface of plain silicon exhibit much reduced diattenuation. On the other hand, a significant amount of the retardance associated with the functionalized polymer mixture is attributed to the thin film effects of the surfaces, i.e., the phase shift on reflection is quite different for s and p polarized light; equivalently, functionalized polymer film exhibits significant retardance [39]. 

The detected amplitudes of backscattered signal contributions under co-polarized and cross-polarized geometries are reported in Figure 13d and Figure 14a. It is observed that the polymer nanostructure maintains the polarization of the incident laser beam better than the silicon substrate. This observation is also highlighted in Figure 14b, where the residual intensity is plotted for the two materials. Again, the conjugated polymers perform better over the silicon substrate at preserving the polarization of the incident light. In fact, the experimental results of Figure 14c,d indicate clearly that the conjugated polymers exhibit both a higher degree of polarization (DOP) and a reduced linear depolarization ratio (LDR) with respect to the silicon substrate.

Experimental results related to the fabrication of cast and annealed Poly (styrene-b-methyl methacrylate) PS-PMMA optical thin films are shown in Figure 15a,b. It can be observed that the annealed thin films of the PS-PMMA blend exhibit higher DOP than the cast PS-PMMA. As a result, the backscattered light preserves its original polarization. This is supported by the fact that the cast film prior to annealing exhibit a pitted topography; meanwhile, as the annealing process proceeds, these pits become continually shallower, and the surface become smoother. This is shown in Figure 7, where the sample surface is exceptionally smooth, lacking distinguishable features, while the presence of a parallel orientation of lamellae is apparent.

Circularly polarized light was detected through the backscattering of the polarized laser beam from different nanofilm depths of the conjugated PS-PMMA blends and only PS-PMMA. The degree of circular polarization (DOCP) for all three samples of different thicknesses, consisting of 120 nm and 180 nm functionalized PS-PMMA with 10% Au, and an 180 nm PS-PMMA (pure) is shown in Figure 16. The observation that 180 nm functionalized PS-PMMA with 10% Au exhibited the highest DOCP with respect to the other two samples, reinforces the knowledge that a mixture of embedded nanoparticles can exhibit higher circularly polarized light (CPL) scattering, while providing enhanced tunability in conjunction with the dielectric medium [69]. Circularly polarized light is central to many photonic technologies, such as metasurfaces, quantum-based optical computing and information, the design of ellipsometry instrumentation, and optical communications [62,63,64,65,66,67]. Indeed, the quantitative detection of circularly polarized light (CPL) is necessary in next-generation high-data optical communication systems and in phase-controlled 3D displays [70].

Angular optical polarimetric scattering measurements from two different thicknesses of annealed PS-b-P2VP diblock thin films copolymers functionalized with Au nanoparticles were performed. The plots of Figure 17a indicate that the annealed conjugated 105 nm PS-P2VP exhibits reduced diattenuation with respect to the 45 nm thick PS-P2VP optical film; the diattenuation was derived from the Mueller matrix (MM)-calculated diattenuation, according to Equation (27). The degree of polarization (DOP) for the two thicknesses of conjugated PS-P2VP was estimated based on Equation (21), and plotted in Figure 17b. It is observed that the annealed 40 nm PS-P2VP containing 6% Au NPs exhibits higher DOP with respect to the annealed 105 nm PS-P2VP conjugated optical nanofilm. This is supported by Figure 17c, where the annealed 40 nm PS-P2VP containing 6% Au NPs exhibits a smaller depolarization coefficient with respect to the annealed 105 nm PS-P2VP conjugated optical nanofilm. It is observed that there is trade-off between DOP and diattenuation.

Since the embedded metal nanoparticle characteristics are strongly dependent upon the electronic structure of the polymer, the fabrication of technologically useful tunable conjugated polymer blends with an optimized refractive index, shape, size, and spatial orientation would lead to the development of new nanoantennas and metasurfaces.

In addition, the knowledge provided in this study would contribute to the development of a high-sensitivity, high-specificity surface plasmon resonance (SPR) imaging systems for directly sensing biological molecules, proteins, DNA, RNA, and antibodies.

## 6. Conclusions

The design, calibration, and development of a near-infrared (NIR) liquid crystal (LC) multifunctional automated optical polarimeter, aimed at the study and characterization of the polarimetric properties of conjugated polymer optical nanofilms, was presented. Detailed calibration procedures confirmed the accuracy and reliability of the designed electro-optical polarimeter. Various nanophotonic conjugated polymer structures were fabricated and characterized in terms of their Mueller matrix and Stokes parameters. In all cases, the backscattered infrared light was studied and related to polarimetric figures-of-merit (FOM).

The outcome of this study indicates that functionalized polymer blends with gold nanoparticles, depending upon their structure and composition, yield polymer nanomaterials with enhanced optical characteristics, by preserving and modulating the polarization of the incident. As a result, these optical conjugated polymer nanofilms are promising candidates for the development of efficient nanoantennas with applications in the control, modulation, and manipulation of light.

## Figures and Tables

**Figure 1 micromachines-14-01132-f001:**
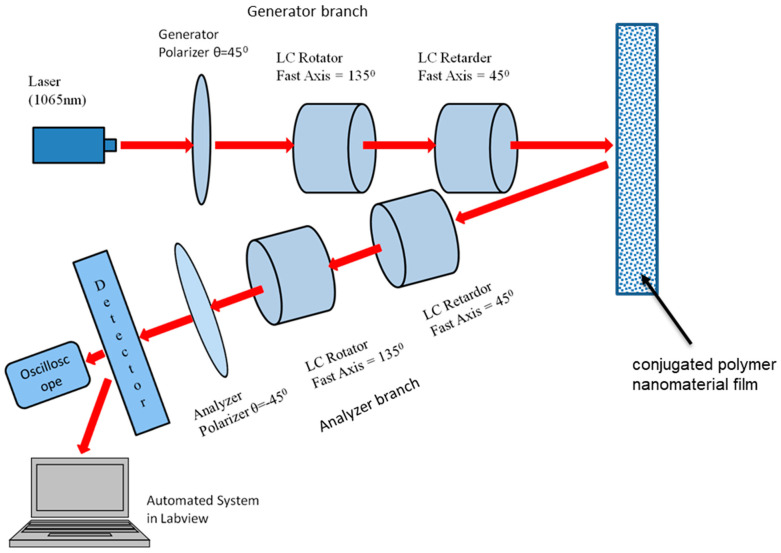
The liquid crystal (LC) multifunctional polarimetric imaging platform.

**Figure 2 micromachines-14-01132-f002:**
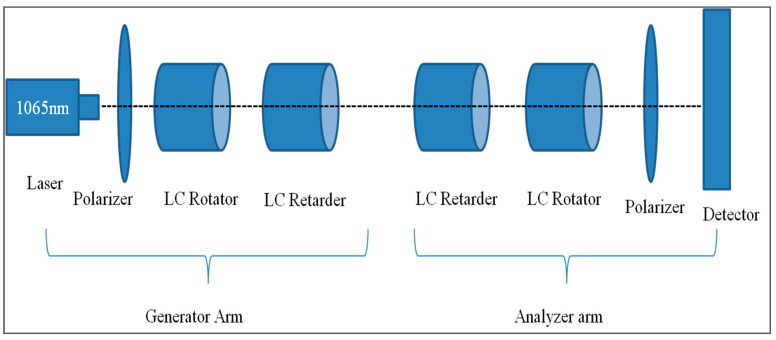
System calibration setup.

**Figure 3 micromachines-14-01132-f003:**
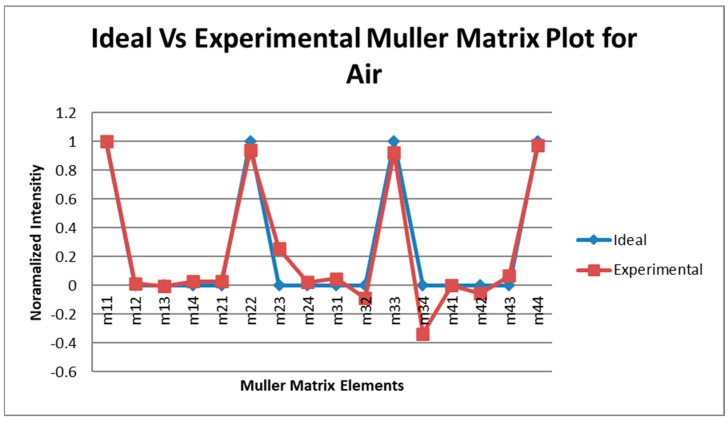
Mueller matrix intensities for the ideal and experimental calibration of air.

**Figure 4 micromachines-14-01132-f004:**
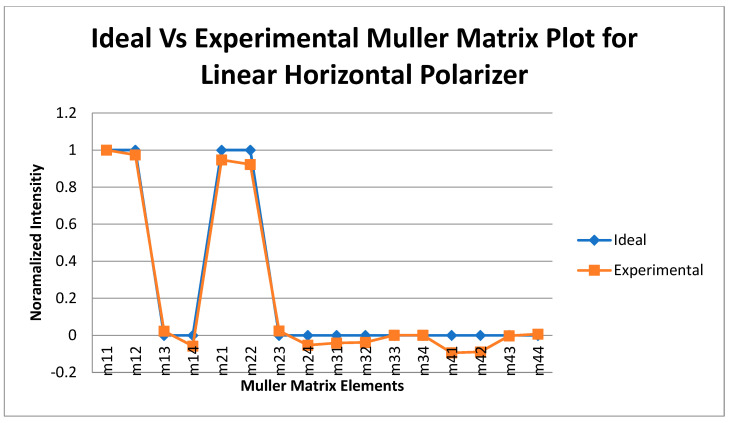
Mueller matrix intensities for the ideal and experimental calibration of a linear horizontal polarizer.

**Figure 5 micromachines-14-01132-f005:**
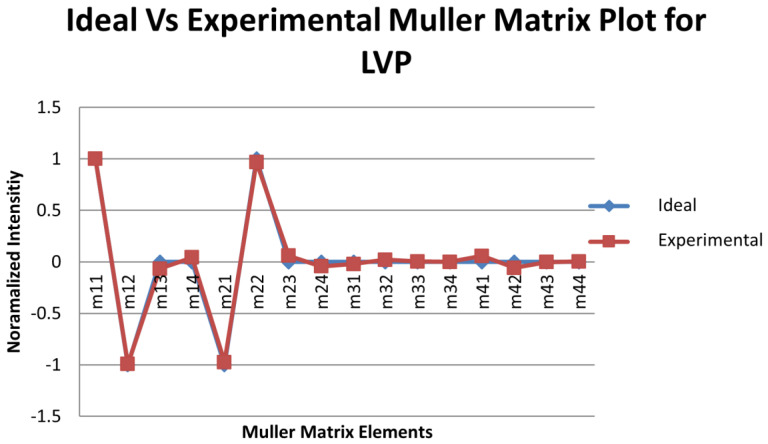
Mueller matrix intensities for the ideal and experimental calibration of a linear vertical polarizer.

**Figure 6 micromachines-14-01132-f006:**
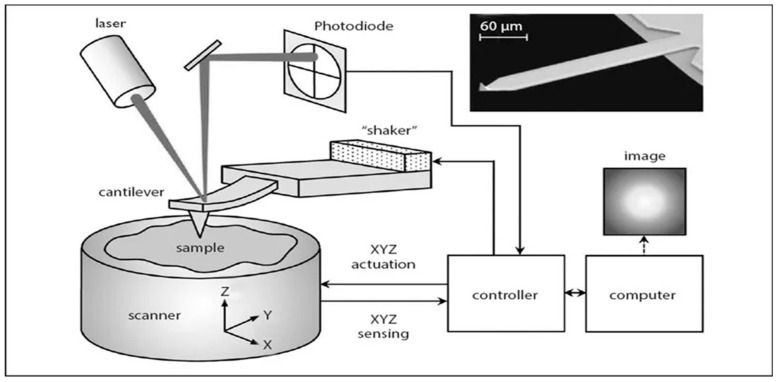
Principles of atomic force microscopy (AFM) [68].

**Figure 7 micromachines-14-01132-f007:**
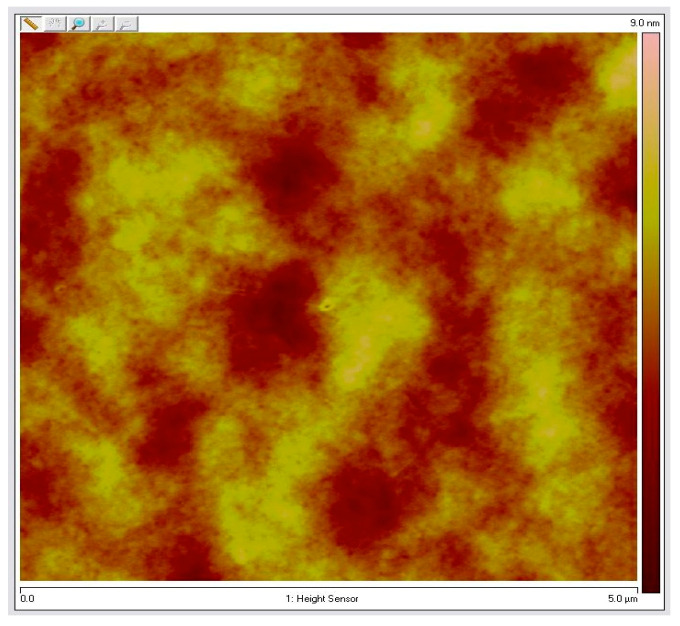
AFM measurement of the PS-PMMA 183 nm over 3 h of oven annealing.

**Figure 8 micromachines-14-01132-f008:**
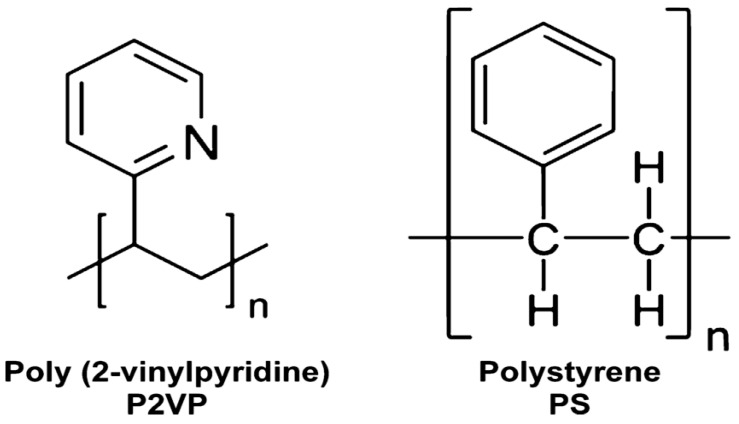
Chemical structure of the P2Vp and PS.

**Figure 9 micromachines-14-01132-f009:**
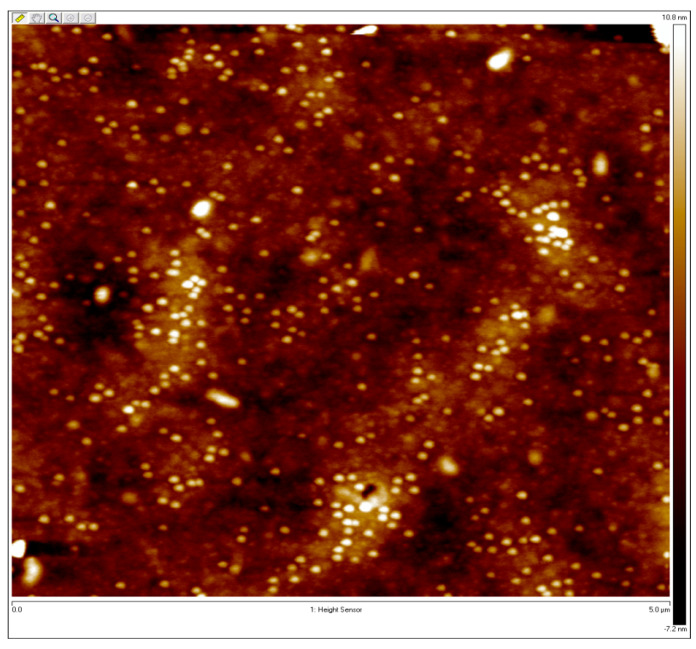
AFM measurement of the PS-P2VP annealed sample containing 6% Au NPs with 40 nm thickness.

**Figure 10 micromachines-14-01132-f010:**
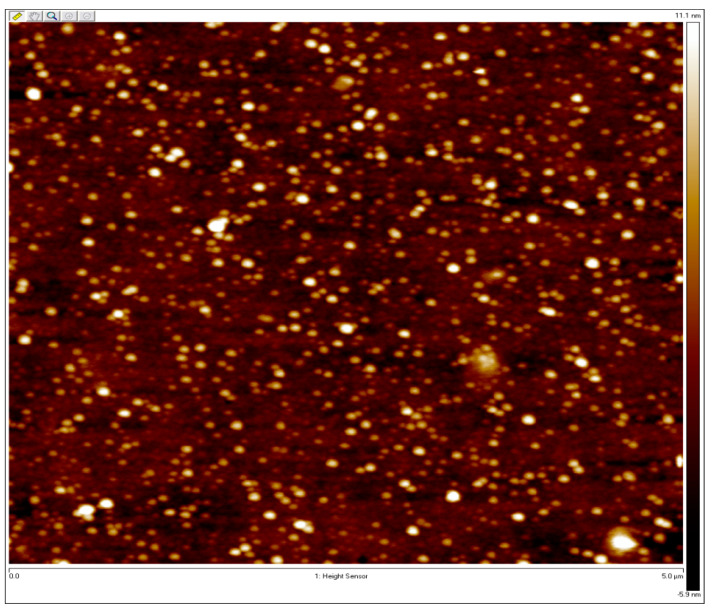
AFM measurement of the PS-P2VP annealed sample containing 6% Au NPs with 105 nm thickness.

**Figure 11 micromachines-14-01132-f011:**
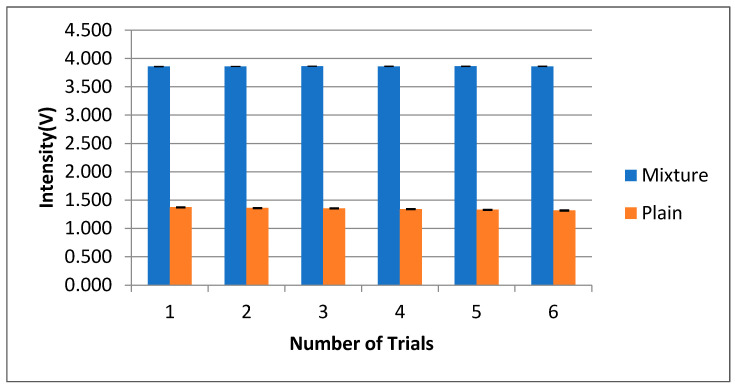
Repeatability experiments under co-polarized geometry.

**Figure 12 micromachines-14-01132-f012:**
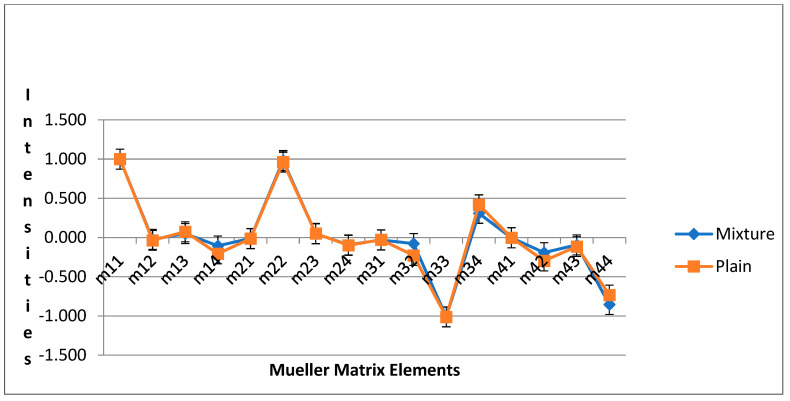
Mueller matrices of conjugated polystyrene (PS) and polybutadiene (PB) blended with Au NPs, and plain silicon substrate.

**Figure 13 micromachines-14-01132-f013:**
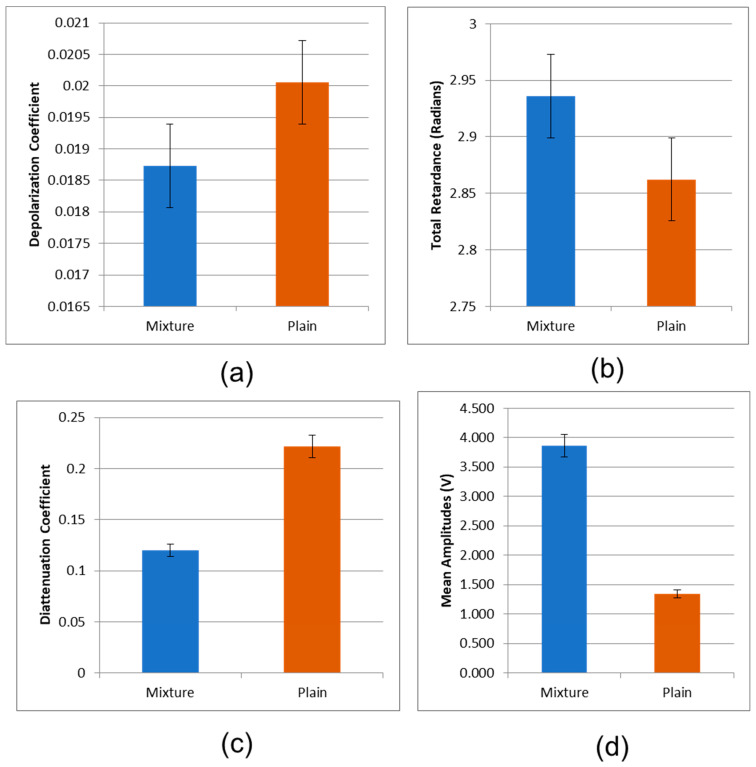
(**a**) Depolarization index plot of mixtures of functionalized polystyrene (PS) and polybutadiene (PB) domains with Au NPs and plain silicon substrate. (**b**) Retardance plot of mixtures of functionalized polystyrene (PS) and polybutadiene (PB) blended with Au NPs and plain silicon substrate. (**c**) Diattenuation plot of thin films of functionalized polystyrene (PS) and polybutadiene (PB) domains with Au NPs and plain silicon substrate. (**d**) Polarized backscattering light intensities from thin films of functionalized polystyrene (PS) and polybutadiene (PB) domains with Au NPs, and plain silicon substrate.

**Figure 14 micromachines-14-01132-f014:**
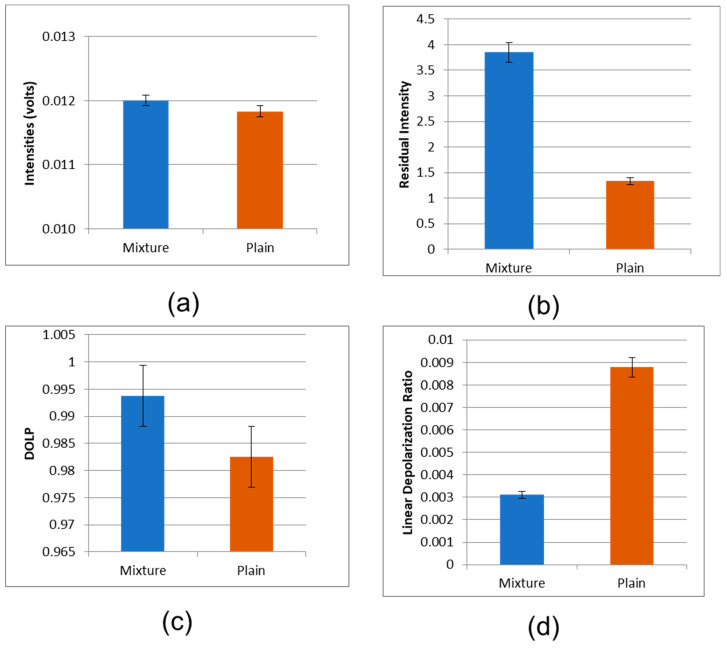
(**a**) Polarized backscattering light intensities from thin films of functionalized polystyrene (PS) and polybutadiene (PB) domains with Au NPs, and plain silicon substrate, under cross-polarized geometry. (**b**) Residual backscattering 1ight intensities from thin films of functionalized polystyrene (PS) and polybutadiene (PB) domains with gold nanoparticles, and plain silicon substrate. (**c**) Degree of linear polarization (DOLP) of backscattering light intensities from thin films of functionalized polystyrene (PS) and polybutadiene (PB) domains with gold nanoparticles, and plain silicon substrate. (**d**) Linear depolarization ratio (LDR) of backscattering light intensities from thin films of functionalized polystyrene (PS) and polybutadiene (PB) domains with gold nanoparticles, and plain silicon substrate.

**Figure 15 micromachines-14-01132-f015:**
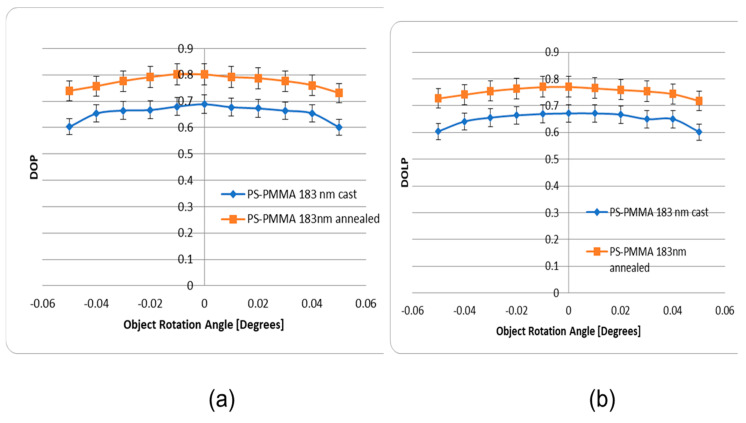
(**a**) Comparison of the DOP of the as-cast and as-annealed 183 nm PS-PMMA sample for different rotation angles of the object. (**b**) Comparison of the DOLP of the as-cast and annealed 183 nm PS-PMMA (no Au NPs) sample for different rotation angles of the object.

**Figure 16 micromachines-14-01132-f016:**
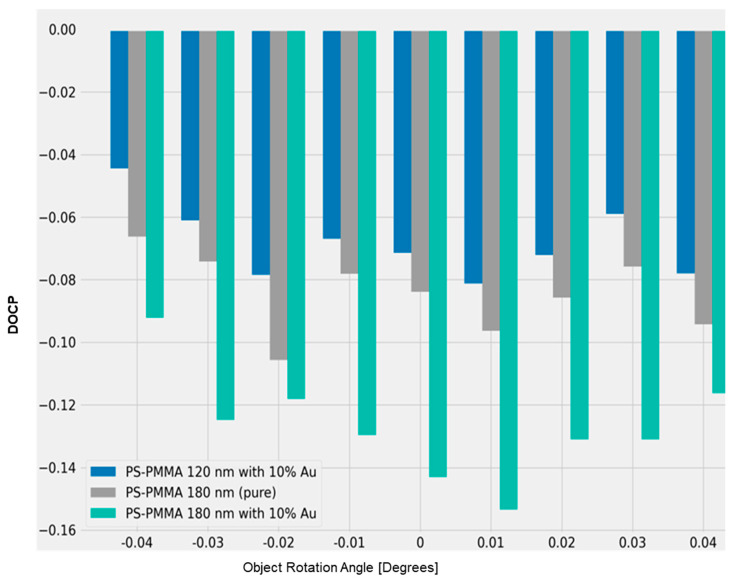
Degree of circular polarization (DOCP) for all three samples with different thicknesses, consisting of 120 nm and 180 nm functionalized PS-PMMA with 10% Au, and an 180 nm PS-PMMA (pure).

**Figure 17 micromachines-14-01132-f017:**
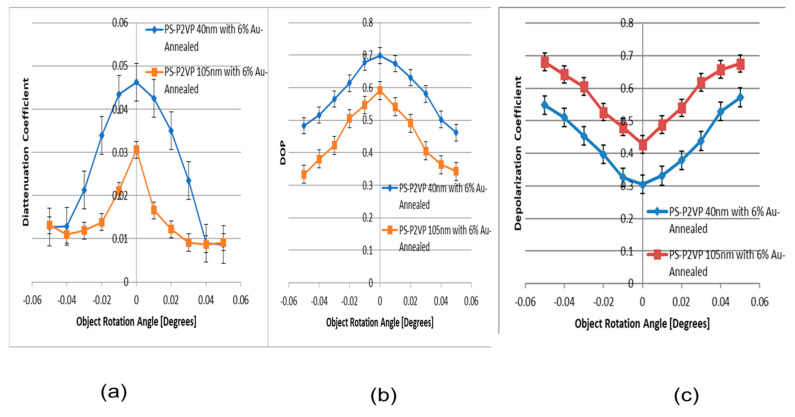
(**a**) Comparison of diattenuation coefficient for the annealed 105 nm PS-P2VP containing 6% Au NPs nm vs. annealed 40 nm PS-P2VP containing 6% Au NPs. (**b**) Comparison of DOP for the annealed 105 nm PS-P2VP containing 6% Au NPs nm vs. annealed 40 nm PS-P2VP containing 6% Au NPs. (**c**) Comparison of depolarization coefficient for the annealed 105 nm PS-P2VP containing 6% Au NPs nm vs. annealed 40 nm PS-P2VP containing 6% Au NPs.

**Table 1 micromachines-14-01132-t001:** Statistical analysis of Mueller matrix for air, LHP, and LVP.

Element	Measured Muller Matrix			Ideal Muller Matrix			Element Variance		
	Air	LHP	LVP	Air	LHP	LVP	Air	LHP	LVP
m11	1.000	1.000	1.000	1	1	1	0.00 × 10^0^	0.00 × 10^0^	0.00 × 10^0^
m12	0.014	0.974	−0.992	0	1	−1	2.07 × 10^−4^	6.84 × 10^−4^	5.79× 10^−5^
m13	−0.005	0.023	−0.066	0	0	0	2.55 × 10^−5^	5.42 × 10^−4^	4.36 × 10^−3^
m14	0.028	−0.058	0.043	0	0	0	8.09 × 10^−4^	3.38 × 10^−3^	1.83 × 10^−3^
m21	0.030	0.947	−0.974	0	1	−1	9.13 × 10^−4^	2.79 × 10^−3^	6.89 × 10^−4^
m22	0.940	0.923	0.968	1	1	1	3.55 × 10^−3^	5.91 × 10^−3^	9.97 × 10^−4^
m23	0.257	0.024	0.060	0	0	0	6.60 × 10^−2^	6.00 × 10^−4^	3.64 × 10^−3^
m24	0.023	−0.053	−0.044	0	0	0	5.43 × 10^−4^	2.82 × 10^−3^	1.97 × 10^−3^
m31	0.048	−0.041	−0.021	0	0	0	2.31 × 10^−3^	1.69 × 10^−3^	4.30 × 10^−4^
m32	−0.089	−0.038	0.020	0	0	0	7.94 × 10^−3^	1.44 × 10^−3^	3.99 × 10^−4^
m33	0.926	0.001	0.003	1	0	0	5.51 × 10^−3^	3.73 × 10^−7^	7.09 × 10^−6^
m34	−0.335	0.001	0.000	0	0	0	1.12 × 10^−1^	6.15 × 10^−7^	1.06 × 10^−13^
m41	0.001	−0.094	0.058	0	0	0	4.46 × 10^−7^	8.81 × 10^−3^	3.37 × 10^−3^
m42	−0.053	−0.089	−0.059	0	0	0	2.83 × 10^−3^	7.86 × 10^−3^	3.46 × 10^−3^
m43	0.067	−0.002	−0.003	0	0	0	4.55 × 10^−3^	2.74 × 10^−6^	9.27 × 10^−6^
m44	0.972	0.007	0.003	1	0	0	7.63 × 10^−4^	4.27 × 10^−5^	7.09× 10^−6^
Standard Deviation							1.14 × 10^−1^	4.78 × 10^−2^	3.64 × 10^−2^

## Data Availability

The data presented in this study are available on request from the corresponding author.

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
