# Peer review of "Next-Generation Reconfigurable Nanoantennas and Polarization of Light"

_micromachines, 2023, doi:10.3390/mi14061132_

Round 1

Reviewer 1 Report

In the manuscript titled “Next Generation Reconfigurable Nanoantennas and Polarization of Light” the authors design a near-infrared liquid crystal polarimeter, and introduce the calibration and stability testing of the polarimeter through experiments. It has been tested that this polarimeter can detect the Mueller matrix elements of air, linear horizontal polarizers, and linear vertical polarizers, and further use the polarimeter to characterize various types of nano-photonic structures experimentally. The experimental results show that the tunable optical properties exhibited by functional polymer nanomaterials have promising optical prospects. The article has some innovation in application, but some expressions and figures in the article are too rough. It is hoped that the author can improve the manuscript further. In summary, I recommend that this manuscript could be published in the Micromachines journal after addressing those issues as follows.

1.      What is the purpose of introducing the optical and physical properties of PC, COC, and ADC materials in Table 1 of the introduction part of the manuscript? Content unrelated to the research object of this paper may reduce the readability for the reader.

2.      At the end of the introduction section of the manuscript, it is mentioned that the characterization of new nano-photonic structures has been achieved in other articles in terms of Mueller matrix and Stokes parameter analysis. What differences and improvements does this work have compared with previously published work?

3.      The experimental methods mentioned in the manuscript lack innovation. This method is highly consistent with the method and experiments shown in the paper “G. C. Giakos, R. H. Picard, P. D. Dao, P. N. Crabtree and P. J. McNicholl, "Polarimetric wavelet phenomenology of space materials," Polarimetric wavelet phenomenology of space materials," 2011 IEEE International Conference on Imaging Systems. The manuscript should highlight the advantages of its own approach in words.

4.      Regarding Figure 23 in the experimental section, could you provide further details on the quantitative analysis of gold film thickness and circular polarization degree?

5.      Some related work on polarization manipulating metasurfaces should be mentioned, such as, eLight 2(1), 23 (2022), Light Sci. & Appl. 10(1), 24 (2021), Phys. Rev. Lett. 130(12), 123801 (2023), and Nat. Commun. 14(1), 1035 (2023).

6.      Some details in the paper should be revised. Firstly, the expressions are inconsistent. The introduction part describes that the working beam of the experiment is 1060nm. Figure 1 in the manuscript shows a liquid crystal multifunctional imaging platform, and the legend sets a 1050nm laser, but the corresponding text explanation is using a 1065-nm laser source. In addition, the markings in Figure 1 should be further enhanced. For ease of reader comprehension, the polarization generation branch and polarization analysis branch should be labeled according to the text description. Apart from this, there are spelling mistakes in the manuscript, such as 'nu' at line 159, which should be 'nm'. The experimental figures in the manuscript are drawn too roughly, like Figure 6. Can the images be made more refined?

Author Response

Reviewer 1

We would like to thank the knowledgeable Reviewer for the constructive reviews aimed at the improving of the overall manuscript quality.

  1. What is the purpose of introducing the optical and physical properties of PC, COC, and ADC materials in Table 1 of the introduction part of the manuscript? Content unrelated to the research object of this paper may reduce the readability for the reader.

Thank you for your valuable suggestion. This Table has been removed.

  1. At the end of the introduction section of the manuscript, it is mentioned that the characterization of new nano-photonic structures has been achieved in other articles in terms of Mueller matrix and Stokes parameter analysis. What differences and improvements does this work have compared with previously published work?

Thank you for the Reviewer’s  insightful question. Starting from our original study on nanophotonic structures consisting of two-different polymer domains functionalized with gold nanoparticles, our research expanded into new domains consisting of cast and annealed Poly (styrene-b-methyl methacrylate) (PS-PMMA) block copolymers, and block copolymers functionalized with gold with gold nanoparticles. The focus of this integrated study is to explore how these materials would lead to reconfigurable photonic structures capable of modulating and controlling polarized light, assessment made using polarimetric Figures-of-Merits (FOM)s.

  1. The experimental methods mentioned in the manuscript lack innovation. This method is highly consistent with the method and experiments shown in the paper “G. C. Giakos, R. H. Picard, P. D. Dao, P. N. Crabtree and P. J. McNicholl, "Polarimetric wavelet phenomenology of space materials," Polarimetric wavelet phenomenology of space materials," 2011 IEEE International Conference on Imaging Systems. The manuscript should highlight the advantages of its own approach in words.

The application of these experimental methods into different domains spanning from Space research to functionalized polymer nanostructures and cancer detection, contribute significantly towards the generation of new knowledge, in addition to the design, fabrication, assessment, and characterization of novel nanophotonic materials (reference to this study), as well as  for remote material inspection (Space materials), and early pathologies detection and monitoring of disease (cancer research). By means of Stokes parameters analysis, it is possible to characterize the backscattering electromagnetic field waves, after interacting with the sample material, while Mueller matrix analysis allow to study the optical and structural characteristics of the sample nanostructured polymers.

  1. Regarding Figure 23 in the experimental section, could you provide further details on the quantitative analysis of gold film thickness and circular polarization degree??

              The observation that 180 nm functionalized PS-PMMA with 10% Au exhibit the highest DOCP with respect to the other two samples, reinforces knowledge that a mixture of embedded nanoparticles can exhibit higher circularly polarized (CP) scattering, while providing enhanced tunability in conjunction with the dielectric medium [70]. Circularly polarized light is central to many photonic technologies, such as metasurfaces, quantum-based op-tical computing and information, design of ellipsometry instrumentation, and optical communications. Indeed, the quantitative detection of circularly polarized light (CPL) is necessary in next-generation big data optical communication systems and in phase-controlled 3-d displays [71].

  1. Some related work on polarization manipulating metasurfaces should be mentioned, such as, eLight 2(1), 23 (2022), Light Sci. & Appl. 10(1), 24 (2021), Phys. Rev. Lett. 130(12), 123801 (2023), and Nat. Commun. 14(1), 1035 (2023).

 Thank you to the Reviewer for providing us with such a wealth of information, which has been added to our References.

  1. Some details in the paper should be revised. Firstly, the expressions are inconsistent. The introduction part describes that the working beam of the experiment is 1060nm. Figure 1 in the manuscript shows a liquid crystal multifunctional imaging platform, and the legend sets a 1050nm laser, but the corresponding text explanation is using a 1065-nm laser source. In addition, the markings in Figure 1 should be further enhanced. For ease of reader comprehension, the polarization generation branch and polarization analysis branch should be labeled according to the text description. Apart from this, there are spelling mistakes in the manuscript, such as 'nu' at line 159, which should be 'nm'. The experimental figures in the manuscript are drawn too roughly, like Figure 6. Can the images be made more refined?

The experimental figures have been revised. Figure 1 has been redesigned so to highlight the intrinsic merit of the instrument and polarimetric technique, unmistakably. The laser source operates at 1065 nm, and the typo error was corrected.

Reviewer 2 Report

In the paper 'Next Generation Reconfigurable Nanoantennas and Polarization of Light' the authors present their recent results about liquid crystal multifunctional automated optical polarimeter. The paper is overall well written, however I strongly believe that the authors should consider the following suggestions before publications:

1. In the introduction part, it is missing the all dielectric-diffractive metasurface with liquid crystals structures;

2. The authors should merge somehow figs.13,14,15,16,17,18,19,20

3. The same holds for figs 21-22 and 24-26

4. I believe that some figures can be placed in the supplementary sections. 

Author Response

  1. In the introduction part, it is missing the all dielectric-diffractive metasurface with liquid crystals structures;

Your valuable suggestion has been addressed by adding this knowledge with pertinent references.

  1. The authors should merge somehow figs.13,14,15,16,17,18,19,20

Thank you for your suggestion. These figures have been merged.

  1. The same holds for figs 21-22 and 24-26

Similarly, we merged those figures.

  1. I believe that some figures can be placed in the supplementary sections. 

Thank you for your valuable suggestion. We opened an Appendix, the so called Appendix A, where figures and Tables reflecting common knowledge or basic principles were placed there.
